# Geometry of inter-areal interactions in mouse visual cortex

**Ramakrishnan Iyer**                                          RAMAKRISHNANIYE@GMAIL.COM
*MindScope Program, Allen Institute, Seattle, WA*

**Joshua H. Siegle**                                          JOSHS@ALLENINSTITUTE.ORG
*MindScope Program, Allen Institute, Seattle, WA*

**Gayathri Mahalingam**                                       GAYUMAHALINGAM@GMAIL.COM
*Allen Institute for Brain Science, Seattle, WA*

**Shawn Olsen**                                               SHAWNO@ALLENINSTITUTE.ORG
*MindScope Program, Allen Institute, Seattle, WA*

**Stefan Mihalas**                                            STEFANM@ALLENINSTITUTE.ORG
*MindScope Program, Allen Institute, Seattle, WA*

**Editors:** Sophia Sanborn, Christian Shewmake, Simone Azeglio, Arianna Di Bernardo, Nina Miolane

## Abstract

The response of a set of neurons in an area is the result of the sensory input, the interaction of the neurons within the area as well as the long range interactions between areas. We aimed to study the relation between interactions among multiple areas, and if they are fixed or dynamic. The structural connectivity provides a substrate for these interactions, but anatomical connectivity is not known in sufficient detail and it only gives us a static picture. Using the Allen Brain Observatory Visual Coding Neuropixels dataset, which includes simultaneous recordings of spiking activity from up to 6 hierarchically organized mouse cortical visual areas, we estimate the functional connectivity between neurons using a linear model of responses to flashed static grating stimuli. We characterize functional connectivity between populations via interaction subspaces. We find that distinct subspaces of a source area mediate interactions with distinct target areas, supporting the notion that cortical areas use distinct channels to communicate. Most importantly, using a piecewise linear model for activity within each trial, we find that these interactions evolve dynamically over tens of milliseconds following a stimulus presentation. Inter-areal subspaces become more aligned with the intra-areal subspaces during epochs in which a feedforward wave of activity propagates through visual cortical areas. When the short-term dynamics are averaged over, we find that the interaction subspaces are stable over multiple stimulus blocks. These findings have important implications for understanding how information flows through biological neural networks composed of interconnected modules, each of which may have a distinct functional specialization.

**Keywords:** mouse visual cortex, inter-areal interactions, geometry, subspace angles, recurrent neural networks, Neuropixels, Allen Brain Observatory, visual coding, neural coding, functional connectivity

## 1. Introduction

The view that cerebral cortex is organized in multiple areas, each of which may carry out different functions, has been around for more than a century (Brodmann, 1909). For primates (Felleman and Van Essen, 1991) as well as for the mouse (Wang et al., 2012; Harris et al., 2019), the visual system is both modular and hierarchical. Different visual areas in mice have distinct retinotopic maps (Zhuang et al., 2017) and functional specialization (Marshel et al., 2011), but are strongly interconnected with each other (Oh et al., 2014; Knox et al., 2018). Understanding how different brain areas work together requires observing activity simultaneously across multiple constituent interacting neuronal populations.

The vast majority of our knowledge of cellular-level interactions between cortical regions comes from experiments in which only two areas were observed at a given time. The Allen Brain Observatory Visual Coding Neuropixels dataset includes simultaneous measurements of spiking activity from all layers of 4-6 cortical visual areas, making it possible to address a wide range of questions about the dynamic routing of information. A recent study (Siegle et al., 2021) using this open dataset revealed that the organization of inter-areal functional connectivity during visual stimulation mirrors the structural connectivity hierarchy.

We analyzed this dataset with a view to understand dynamics of interactions between *six* mouse visual cortical areas when a simple visual stimulus is presented. We used a linear model (with L1-regularization) which predicts target neuron activities using a linear combination of the activities of all simultaneously recorded neurons from multiple brain regions and a stimulus-dependent contribution. Using this model, we obtain interaction matrices between pairs of visual cortical areas. We quantify the degree of alignment between different interaction subspaces using subspace angles. Consistent with the results of Semedo et al. (2019), we find that intra- and inter-areal interactions are distinct for each visual cortical area in mouse.

A unique contribution of our study is the relation between interaction subspaces across multiple visual areas. We find that the inter-areal interactions among visual areas are more aligned than the intra-areal interactions. We also find that the interaction subspaces are dynamic, with rapid changes over ~50ms after a stimulus onset. However, when the short-term dynamics are averaged over, the interaction subspaces are stable over stimulus blocks which are $> 30$ minutes apart. This creates an interesting comparison with the recent study in Srinath et al. (2021) which found that attention does not modify the communication subspace. Our results could help future studies understand the dynamics of formation of representations across areas when the stimulus is task relevant. Details regarding relation to previous work have been provided in Appendix A1.

## 2. Results

### 2.1. Peer prediction model captures interactions between neurons in different brain regions

We used the Allen Brain Observatory Visual Coding Neuropixels dataset (https://portal.brain-map.org/explore/circuits/visual-coding-neuropixels). It contains simultaneous recordings of neuronal responses to a variety of visual stimuli from six visual cortical areas (VISp, VISl, VISal, VISrl, VISpm and VISam) in addition to thalamic regions LP,

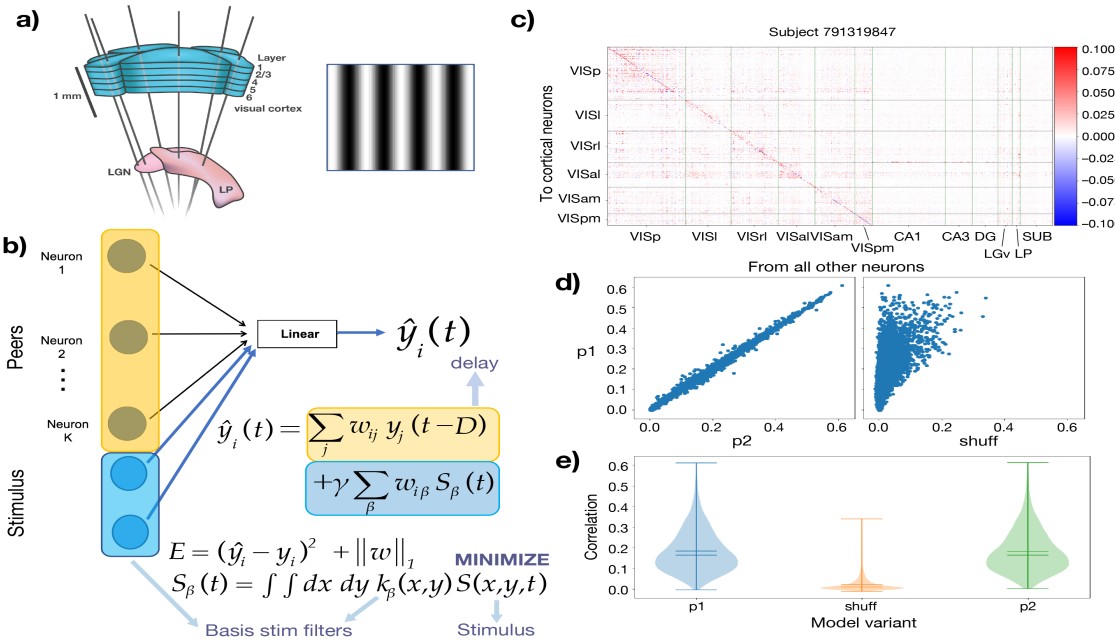

Figure 1: Overview of model for interactions. a) Figure showing six Neuropixels probes targeting cortex and thalamus with static grating stimuli analyzed in this study. b) Schematic of linear model with L1 regularization for single neuron responses using stimulus and peer terms. Stimulus contribution is evaluated using a bank of 120 basis Gabor filters (see Appendix A2). c) Peer coupling coefficients from units in all regions (columns) on to units in all visual cortical regions (rows) from an example session. Vertical and horizontal lines demarcate units within a given source and target region. d) Scatter plot comparing correlations between model predictions and test data for all visual cortical units ($\approx 5700$ units) for three model variants. In all cases, the vertical axis represents test correlations from the model trained on the first half of data (p1). The horizontal axis shows correlation from model variants. Left : model trained on the second half of data (p2) and tested against the first half; Right: model trained using shuffled data (see Appendix A2). e) Distribution of correlations between model predictions and test data for all units in six visual cortical regions for the three model variants.

dLGN and hippocampus in awake head-fixed mice on a wheel using Neuropixels probes. Further details can be found in Siegle et al. (2021).

We analyzed responses to flashed static grating stimuli (Figure 1(a)). We focus on the flashed gratings as they are relatively simple stimuli which still allow us to reveal a temporal dependence of interactions. Recent work (Das and Fiete, 2020) has shown that functional interactions can be more easily discovered by analyzing stimulus-driven responses of the network than just from spontaneous activity. The stimulus set comprised of gratings with six orientations (0, 30, 60, 90, 120, 150) degrees, five spatial frequencies (0.02, 0.04, 0.08, 0.16, 0.32) cpd and four phases (0, 45, 90, 135) degrees. Each grating was flashed for 250 ms approximately 50 times for a total of 25 minutes of static grating stimulus presentations (with gray screens of the same duration randomly interleaved approximately every 30 presentations). Stimuli were presented in three epochs of 9, 8 and 8 minutes respectively with 22 and 15.5 minutes gaps between the blocks. We selected neurons based on three quality metrics which ensured that units were relatively stable across the entirety of each session ($\approx 90$ minutes)

and had low levels of spike train contamination (see Appendix A2). We chose 19 sessions that contained at least 10 units in five out of six visual cortical areas. This gave us a total of 5686 units across the 19 chosen sessions. We provide a breakdown of the number of units per visual area for each of the 19 sessions analyzed in Appendix A3. We collected together spike-times for each neuron in a session across all three epochs and binned them in 10 ms bins.

To investigate the nature of interactions between mouse visual cortical areas, we developed a generalized linear model to predict single neuron activity on each trial of stimulusFigure 1(b), see Appendix A2). Figure 1(c) shows a matrix of coupling coefficients from all neurons to all visual cortical neurons estimated from the model for an example session. While individual neurons' own spiking history provides some of the largest coefficients as seen from the diagonal terms in the matrix, the model also captures strong long-range influences from neurons in multiple other brain regions. Most of the largest coefficients come from other visual cortical and thalamic regions which are well-known to be involved in visual stimulus processing. We also show coefficients from stimulus contributions in Appendix A2.2.

We constructed different model variants labeled (p1, p2, shuff) to ascertain the robustness of our estimates for the coupling coefficients and subsequent analyses. These models differ in the training data that is used to estimate the weights (see Appendix A2). We evaluated model performance using the correlation between model prediction and test data. Figure 1(d) shows scatter plots of correlations between the base model (p1) and model variants (p2; left plot) and (shuff; right plot) for all 5686 visual cortical neurons analyzed. The left plot shows that the model is robust to choice of training data and is not sensitive to the particular stimulus epochs chosen to train the model. The right plot shows that the base model outperforms the shuffled model indicating that it is capable of capturing meaningful stimulus-specific patterns of interactions. Correlations range from slightly negative to as high as 0.6 for a few neurons, with medians 0.163 (0.163) respectively for p1 (p2) and 0.009 for shuffled data, comparable to values reported in de Vries et al. (2020). Figure 1(e) shows distributions of correlations for the three model variants. Neurons having high correlation even with shuffled data are likely high-firing neurons (possibly interneurons) that are agnostic to stimulus.

## 2.2. Cortical areas interact via distinct subspaces

It was shown in Semedo et al. (2019) that cortical areas V1 and V2 in macaques interact via a communication subspace. V2 activity was related to a small subset of population activity patterns in V1 and these patterns were distinct from shared V1 fluctuations which were most predictive of V1 activity. Motivated by their findings, we used the estimated functional interaction matrices from our model to investigate how distinct are interactions between pairs of mouse visual cortical areas. Figure 2(a) shows a schematic of this idea.

To probe the geometry of functional interactions between pairs of visual cortical areas, we calculated the subspace angles $\theta(S, T_1, T_2)$ between subspaces of estimated interaction matrices $(M_1, M_2)$ from a source area $S$ to pairs of target areas $(T_1, T_2)$ (see Appendix A2). The subspace angle generalizes the notion of angle between a vector and 2D plane in 3 dimensions. It measures the degree of alignment between two subspaces in a higher dimensional space and varies from 0 degrees (indicating perfect alignment) to 90 degrees (indicating orthogonality between the two subspaces). Intuitively, the subspace angles capture the angles between the first few most important modes of interaction between $M_1$ and $M_2$.

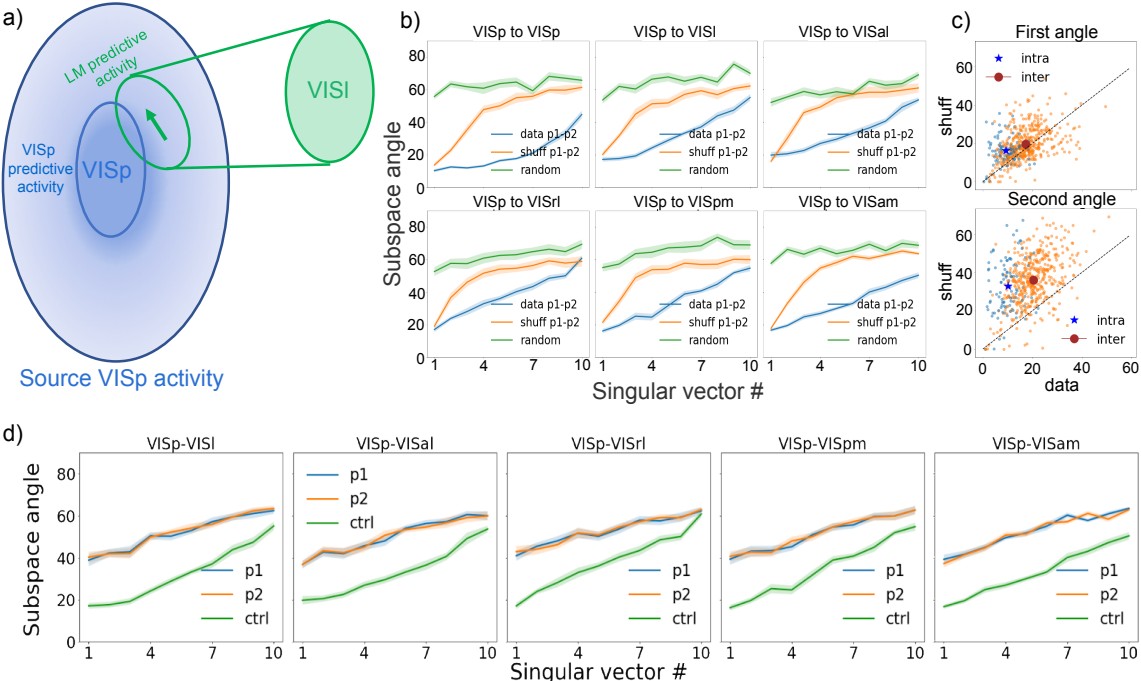

Figure 2: Distinct subspaces of source areas mediate intra- and inter-areal interactions. a) Schematic of the idea showing that distinct subspaces in VISp mediate interactions within VISp and outside with VISl. b) Subspace angles (see Appendix A2) provide a measure of how distinct interaction subspaces between pairs of areas are aligned. Subspace angles for interaction matrices from VISp to six target regions estimated using first and second halves of activity for original data (blue), shuffled data (orange) and a fully randomized condition (see Appendix A2). Solid lines represent session averages; shaded regions show std error of mean (sem). c) Scatter plots of first (top) and second (bottom) subspace angles from original data (horizontal axis) vs shuffled data (vertical axis) for intra-areal (blue) and inter-areal (orange) interactions. Angles from original data are significantly lower than those obtained using shuffled data (Wilcoxon signed-rank test; pvals : $(1.8 \times 10^{-13}, 3.7 \times 10^{-9}, 4.4 \times 10^{-18}, 7.3 \times 10^{-67})$ respectively for first intra-, first inter-, second intra- and second inter-areal interaction angles.) d) Subspace angles from a fixed source region (VISp shown here) to (VISp, T) where T is one of five higher visual areas [T : VISl, VISal, VISrl, VISpm, VISam] for model variants p1 (blue) and p2 (orange). Angles are significantly higher than inter-areal interaction angles (green) estimated from activities in different stimulus blocks (blue lines in Figure 2(b)) as well the shuffle controls (see Appendix A5).

We first verified that the subspace angles obtained from our model provide a robust measure of interaction geometry by comparing the angles between matrices for a source and target area pair obtained from model variants (p1, p2) containing recordings from different stimulus blocks for both original and shuffled data. Figure 2(b) shows angles between matrices from source $S$ (VISp here) to all visual cortical areas averaged across all sessions. Figure 2(c) shows a scatter plot comparing first (top row) and second (bottom row) angles from original and shuffled data for all possible pairs of source and target areas across all sessions, both for intra-areal (blue) and inter-areal (orange) interactions. The angles from shuffled data are significantly higher than those from original data coming from different stimulus blocks. This tells us that the change in the interaction subspaces across stimulus blocks together with the intrinsic variability in estimating the subspaces is small compared to the shuffle

condition. It is interesting to note here that the first angles from original and shuffled data are quite close to each other. This is likely due to the fact that in both cases, the binned activities have been aligned to stimulus onset.

Next, we investigated the subspace angles between intra-areal and inter-areal interactions - namely recurrent interactions between neurons within a source area and the interactions of the same group of neurons with those in other areas. Using the notation introduced above, we are interested in the angles $\theta(S, S, T)$ when $M_1$ represents recurrent interactions within $S$ and $M_2$ represents interactions between $S$ and an external target area $T$.

The angles between these subspaces are significantly higher than the angles between inter-areal subspaces estimated for different stimulus blocks (Figure 2(d)) and they are also significantly higher than shuffle control (Appendix A5). These results are consistent with the findings in macaque for V1-V2 interactions (Semedo et al., 2019) and extend beyond VISp (V1) to all higher visual cortical areas in mouse. We also computed subspace angles after estimating interaction matrices using the approach of Semedo et al. (2019). We show in Appendix A5 that this latter method produces qualitatively similar results.

We then asked how aligned are subspace interactions from a source cortical area to two external cortical areas. Are external-external interactions from a source population of neurons mediated via distinct modes (Figure 3(a))? We are thus interested in the cases when $M_1$ and $M_2$ correspond to interactions between source area $S$ and external target areas $T_1$ and $T_2$. We show in Appendix A5 that angles $\theta(S, T_1, T_2)$ between external-external interactions are again higher than shuffle controls.

### 2.3. Internal-external interaction subspaces are less aligned than external-external interaction subspaces

Having established that cortical areas interact via distinct subspaces, we next investigated if there are differences between internal-external interactions and external-external interactions from a source area to different target areas. Specifically, we compared the subspace angles $\theta(S, S, T_1)$ and $\theta(S, T_1, T_2)$. We use internal to refer to recurrent interactions within an area and external to refer to interactions from a source area to an external target.

Figure 3(b) shows the first (top) and second (bottom) subspace angles from a source area to pairs of target areas averaged across all sessions. For each of the six visual cortical areas, the angles between internal-external interaction subspaces are higher than the angles between external-external interaction subspaces. This is particularly prominent when the source area is VISrl - an area that is known to be weakly responsive to visual stimuli compared with other visual cortical areas and has strong connections to somatosensory areas (Knox et al., 2018).

Figure 3(c) shows the comparison between distributions of the internal-external and external-external first (top) and second (bottom) angles from all sessions. For each pair of distributions (except for the first angle for source VISp), the internal-external angles are significantly higher than the external-external angles (Benjamini-Hochberg FDR-corrected p-values for the Mann-Whitney U test for six cortical areas respectively: $(0.128, 0.031, 0.021, 6.69 \times 10^{-8}, 0.001, 0.003)$ for the first angles and $(0.031, 6.56 \times 10^{-7}, 1.04 \times 10^{-5}, 2.53 \times 10^{-9}, 1.82 \times 10^{-7}, 8.76 \times 10^{-8})$ for the second angles).

Do similar patterns of interactions exist within task-trained artificial neural networks? We have shown that intra-areal interaction subspaces are not well aligned with the inter-areal

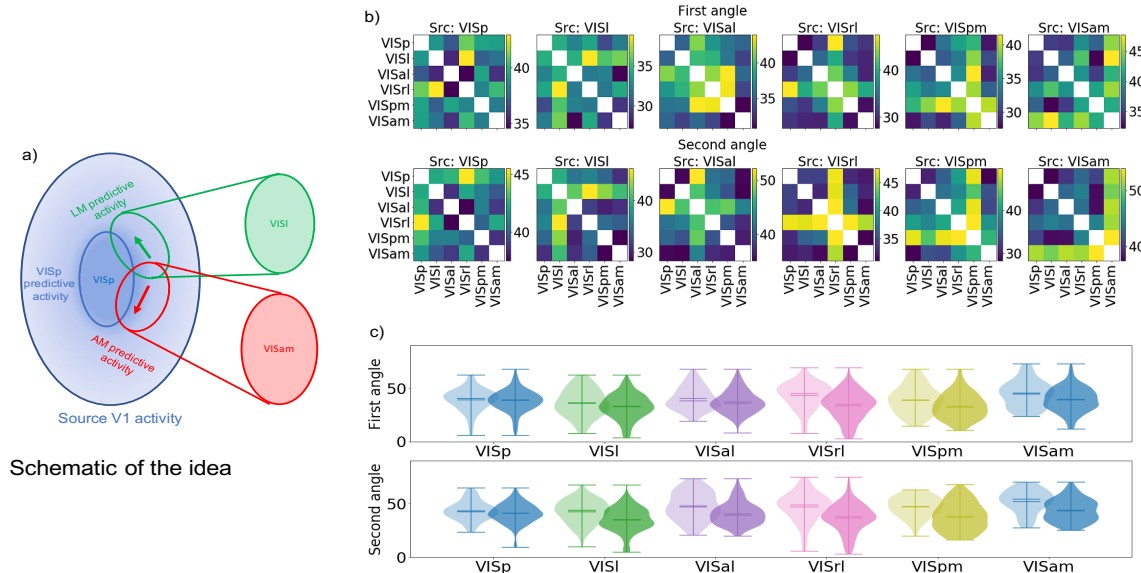

Figure 3: Internal-external interaction subspaces are less aligned than external-external interaction subspaces. a) Schematic of the idea showing that distinct subspaces in VISp mediate interactions with external targets VISl and VISam. b) Heatmaps of first (top row) and second (bottom row) subspace angles between all possible subspaces within each source cortical area. Each pixel represents the subspace angle between $(S,T_1)$ and $(S,T_2)$ where $S$ is the source area and each (row,column) corresponds to $(T_1, T_2)$. The heatmaps are symmetric corresponding to exchange of the $(T_1, T_2)$ labels. Note that the colorbars are on different scales. c) Distribution of internal-external and external-external angles (top; first angle, bottom; second angle). Each pair of violinplots with different shades of the same color represents internal-external (light shade) and external-external (dark shade) angles respectively.

interaction subspaces (Figure 3). To allow for such a possibility, the network is likely to require interactions between neurons within an area. Therefore we analyzed interactions between the representation modules in PredNet (Lotter et al., 2016, 2020) - a deep convolutional recurrent neural network that was trained for next-frame video prediction. PredNet's architecture was inspired by principles of predictive coding from neuroscience (Rao and Ballard, 1999), but it includes both feedback from higher areas as well as lateral connection on top of the feedforward connections.

We provide detailed results of our analysis in Appendix A6. Briefly, we 'probed' the activations of a subset of neurons within representation layers $R_1$, $R_3$ and $R_3$ in PredNet in response to video sequences. These are the layers that have explicit recurrent connections. We used the peer coupling model to estimate internal-external and external-external interaction subspace angles between these modules. In contrast to the biological network, our analysis failed to reveal significant differences between internal-external and external-external interactions within PredNet.

## 2.4. Subspaces exhibit greater alignment during the wave of activity elicited by flashed stimuli

Next we investigated if the degree of alignment between interaction subspaces changes dynamically over the course of tens of milliseconds within a single trial of stimulus presentation. We divided activities evoked over the course of a 250ms stimulus presentation into five

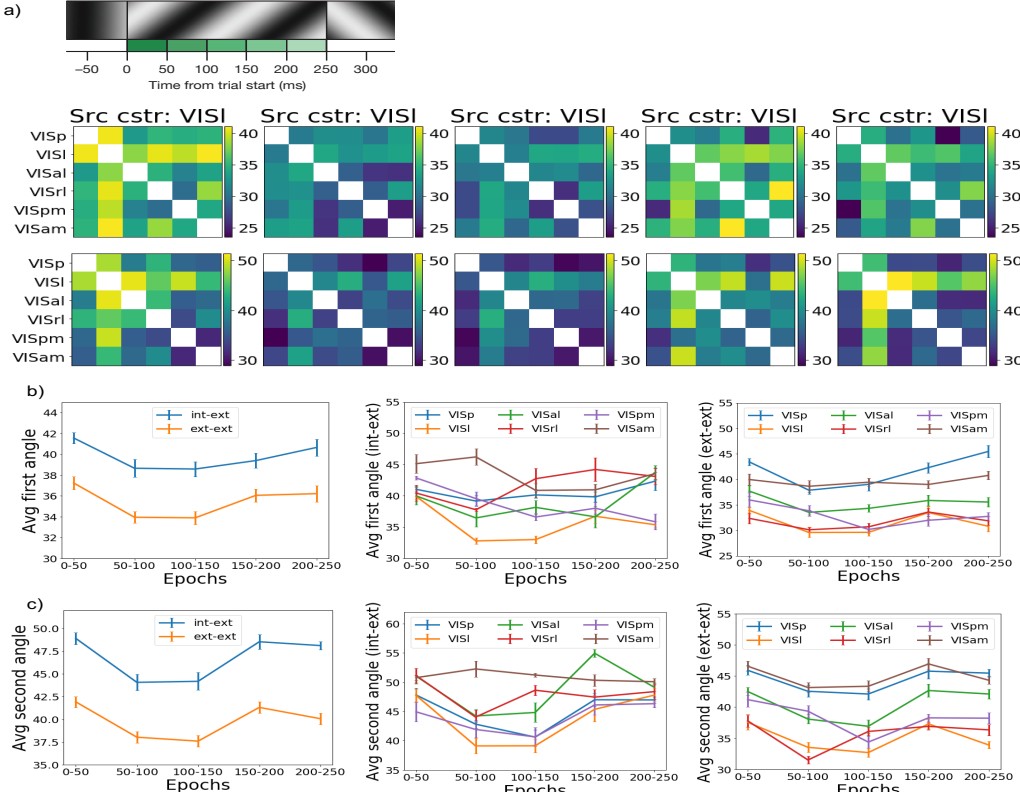

Figure 4: Subspaces rotate within single trials and become more aligned after stimulus onset. a) Top: Schematic showing division of 50ms epochs within a single trial from stimulus onset. Bottom first row: Heatmap of first angle from a source cortical area (VISl shown here) to all possible target cortical area pairs for five successive 50 ms epochs (columns) within 250 ms trials. Bottom second row: Second angles. Angles were computed by estimating interaction matrices using activity within each 50 ms epoch. Note that although subspaces become more aligned, internal-external subspace pairs are still more orthogonal than external-external subspace pairs within each epoch. b) left) Average of all combinations of internal-external (blue) and external-external (orange) first angles for the five 50 ms epochs showing subspace alignment between 50-150 ms after stimulus onset, middle) average of all internal-external first angles across five 50 ms epochs separated by source cortical area and right) average of all external-external first angles across five 50 ms epochs separated by source cortical area. c) Same as in b) but for second angles.

successive epochs of 50 ms each from the time of stimulus onset (Figure 4(a) top). Using the peer-prediction model, we estimated interaction matrices between cortical areas within each 50ms epoch and computed the subspace angles from a source area to all possible pairs of target areas. Our analysis reveals that rather than being static, the degree of alignment between interaction subspaces changes within a single trial of stimulus presentation.

Heatmaps in Figure 4(a) middle and bottom rows respectively show the first and second subspace angles from a source area (VISl shown here) to a pair of target cortical areas for each 50 ms epoch. It can be seen that the angle between interaction subspaces becomes smaller during the (50-100ms) and (100-150ms) epochs before gradually becoming similar to the (0-50ms) epoch. This holds true for each of the six visual cortical areas (Appendix A5.1).

For both internal-external and external-interactions, the angles during the (50-100ms) and (100-150ms) epochs are significantly smaller than the angle during the (0-50ms) epoch ((Benjamini-Hochberg FDR-corrected p-values for Wilcoxon signed-rank test comparing average angles for epochs 2-5 with epoch 1; first angle p-values : $(1.00\times10^{-9}, 1.00\times10^{-9}, 5.69\times 10^{-4}, 3.15\times10^{-2})$, second angle p-values : $(2.65\times10^{-12}, 1.27\times10^{-13}, 1.82\times10^{-1}, 1.66\times10^{-4})$)). Although the angles decrease overall, the internal-external angles still remain higher than the external-external angles within each epoch. This can be seen in the first columns of Figure 4(b,c) which respectively show the first and second angles averaged across all sessions and areas for internal-external (blue) and external-external (orange) interactions.

Session-averaged first and second angles for each source visual cortical area are shown respectively in Figure 4(b,c) middle column (internal-external angles) and last column (external-external angles). These reveal a rich variety of dynamics depending on the specific source area. For example, VISam, which is at the top of the visual hierarchy (Harris et al., 2019; Siegle et al., 2021) does not show the same decrease in intra- and inter-areal interaction subspace angles as areas lower in the hierarchy. VISrl angles rebound faster than the other areas, decreasing during the 50-100ms epoch and increasing again during the 100-150ms epoch.

One important question is if this observed change is the result of changes in firing rates within these intervals. To address this, we used a stimulus shuffle (see Appendix A2 and Figure 2, model variant shuff). The same shuffle, which preserves the temporal structure of the average firing rate of the neurons was used, but it was applied to neuronal activities separated into 50ms time intervals. We applied the same method to compute the subspace angles and their relative alignment for the shuffled data, and we subtracted the shuffle control from all the angles computed. We find that the results are unchanged when the shuffled angles are subtracted, with significant alignment between the inter- and intra-areal subspaces 50-150 ms after the stimulus onset compared to 0-50ms (see Appendix A5.1).

## 3. Discussion

Our study confirms the distinction between the intra- and inter-areal communication subspaces characterized by Semedo et al. (Semedo et al., 2019) in the visual cortex of a different model species (mouse vs primate) and with a slightly different method (see (Appendix A5.3) for a more direct comparison). Our study's contributions are that: 1) We observe that the distinction between intra- and inter-areal interaction subspaces generalizes across all six visual cortical areas, and that the subspaces are stable across stimulus blocks. 2) While all the subspaces are distinct, the intra-areal interaction subspaces are significantly less aligned than the inter-areal interaction subspaces. 3) Most importantly, we observe that the interaction subspaces are dynamic, and can rotate within 50ms following a stimulus onset. However, when averaging over the short-time dynamics, the interaction subspaces are stable over stimulus blocks > 30 minutes apart.

There are multiple limitations of our study. While it would be most interesting to analyze the dynamics of subspaces across task conditions, the available data is limited to passive viewing. It would be interesting to compare the dynamics of interaction subspaces with other methods (e.g. communication by coherence). Another limitation is that we characterized the interaction subspaces only among visual areas, and interactions with other modules/non-visual areas might be different. The biggest limitation is that we do not have

a mechanistic understanding of the circuit structure which causes such a misalignment of inter- and intra-areal subspaces, or the mechanism which causes them to rotate and be better aligned following the stimulus onset. We hoped to be able to peek inside the black box by using an existing artificial neural network and performing experiments by modifying its connectivity structure and seeing when different features of the biologically observed code are lost. However, we could not run such analyses since the network we analyzed (PredNet) did not reproduce the main features observed in biology to begin with.

One way to interpret our results is that in biological networks, individual areas have inter-areal dynamics which is not well aligned with the intra-areal dynamics, but this alignment becomes better for a short period of time following a stimulus onset (i.e. cortical areas communicate what they are doing to other areas only when needed). This communication principle seems different from one artificial recurrent neural network that we looked at. Our study opens up questions about the importance of such a communication principle in more complex tasks in biological and artificial neural networks.

## Acknowledgments

We wish to thank the Allen Institute founder, Paul G. Allen, for his vision, encouragement and support. Stefan Mihalas was in part supported by NIH grants 1RF1DA055669-01 and 1R01EB029813-01.

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

## Appendix: Geometry of inter-areal interactions in mouse visual cortex

### Appendix A1. Expanded description of the relation to previous work

Many studies have investigated inter-areal interactions, mostly between *two* different brain regions. Interactions have been characterized using multiple techniques; a prevalent method consists of recording LFPs in two different areas (Gregoriou et al., 2009; Salazar et al., 2012; Bosman et al., 2012; Jia et al., 2013; Bastos et al., 2015). A corpus of studies focused on understanding the frequency dependence of the signals led to the proposal of communication through coherence (Fries, 2015; Kohn et al., 2020). However, it is not yet clear how the oscillations relate to the formation of specific representations of individual cells needed for image-dependent tasks such as object classification. Other methods include the study of pairwise correlations between spiking activity of neurons in different areas (Siegle et al., 2021; Jia et al., 2013; Reid and Alonso, 1995; Nowak et al., 1999; Ruff and Cohen, 2016), or relating spiking activity of a single neuron in one area and a population in another (Truccolo et al., 2010; Zandvakili and Kohn, 2015). These have led to important insights into the nature and dependence of interactions on factors such as stimulus drive and task demands.

With the advent of large scale simultaneous recordings from multiple brain regions, studies have begun investigating relationships between multiple neuronal spike trains in different areas using multi-variate statistical methods such as multivariate linear regression, canonical correlation analysis (CCA) and their variants (Kaufman et al., 2014; Semedo et al., 2019; Ruff and Cohen, 2019; Ames and Churchland, 2019; Veuthey et al., 2020) (see Semedo et al. (2020) for a review of methods and related studies). Of particular interest to us is the study by Semedo et al. (Semedo et al., 2019) on interactions between cortical areas V1 and V2 in macaques. Using factor analysis and reduced rank regression for dimensionality reduction, they relate trial-to-trial fluctuations between neuronal populations in these areas to show that V1-V2 interactions are mediated via a communication subspace.

To characterize interactions between *six* mouse visual cortical areas with a slightly different approach, we used a linear model (with L1-regularization) which predicts target neuron activities using a linear combination of the activities of all simultaneously recorded neurons from multiple brain regions and a stimulus-dependent contribution. Variants of such *peer-prediction* models have been previously used to characterize interactions within populations (Harris et al., 2003; Pillow et al., 2008) and to study inter-areal interactions (Truccolo et al., 2010; Zandvakili and Kohn, 2015).

Using this model, we obtain interaction matrices between pairs of visual cortical areas. We quantify the degree of alignment between different interaction subspaces using subspace angles. Consistent with the results of Semedo et al. (2019), we find that intra- and inter-areal interactions are distinct for each visual cortical area in mouse. A unique contribution of our study is the relation between interaction subspaces across multiple visual areas. We find that the inter-areal interactions among visual areas are more aligned than the intra-areal interactions.

We also find that the interaction subspaces are dynamic, with rapid changes over 50ms after a stimulus onset. However, when the short-term dynamics are averaged over, the interaction subspaces are stable over stimulus blocks which are > 30 minutes apart. This creates an interesting comparison with the recent study in Srinath et al. (2021) which found that attention does not modify the communication subspace.

## Appendix A2. Methods

### A2.1. Experimental methods overview

Experimental methods are described in detail in Siegle et al. (2021). Mice were head-fixed with the right eye placed 15 cm from a visual stimulus monitor, and were free to run on a rotating disk throughout the experiment. Neuropixels probes (Jun et al., 2017) were inserted into each cortical area, as well as underlying subcortical regions, such as hippocampus and thalamus. Data was acquired with the Open Ephys GUI (Siegle et al., 2017), and spikes were extracted offline using Kilosort2 (Stringer et al., 2019). Units found by Kilosort2 were selected for further analysis based on three quality metrics thresholds: presence ratio $> 0.95$, ISI violations $< 0.5$, amplitude cutoff $< 0.1$ (see Siegle et al. (2021) for definitions). In total, we analyzed 5686 neurons from 19 mice.

### A2.2. Peer prediction model for interactions

The instantaneous firing rate $y_i(t)$ of an individual neuron was predicted in time-bins of size 10 ms as the weighted sum of a peer-prediction term and a stimulus-dependent term as follows,

$$\hat{y}_i(t) = \sum_j w_{ij} y_j(t - D) + \gamma \sum_\beta w_{i\beta} \int dx \int dy \, k_\beta(x, y) S(x, y, t) \qquad (A1)$$

where $k_\beta(x, y)$ represents a basis set of 120 gabor filters and $S(x, y, t)$ represents the static grating stimulus. Gabor filters were parameterized using static grating parameters (6 orientations, 5 spatial frequencies, 4 phases) used in the experiments and $D$ represents a delay that allows us to include the spiking history of individual neurons. We chose $D = 10$ ms for our model.

Coefficients $w_{ij}$ and $w_{i\beta}$ were fit using linear regression with an $L_1$ regularization term $\alpha_\gamma ||w||_1$. The regularization term constrains the number of non-zero coefficients in the model and explains away contributions from neurons which may themselves be highly correlated. $\gamma$ controls the relative contributions of peer activity and stimulus. For a fixed value of $\gamma$, the hyper-parameter $\alpha_\gamma$ was optimized for via 10-fold cross validation on training data. $\gamma$ itself was varied on a grid with $\gamma = (0.0, 0.01, 0.5, 0.1, 1.0, 5.0, 10.0, 50.0)$ and was chosen to be the one that maximized the correlation between model prediction and test data. We used scikit-learn's LassoCV method for fitting. Having done this for a subset of the 19 sessions, we noticed that the specific choice of $\gamma$ did not lead to significant differences in the correlation between model prediction and test data. So without loss of generality, we fixed $\gamma = 1$ for all results reported here.

**Model variants** : We constructed different model variants to ascertain the robustness of estimates for coupling coefficients and subsequent analyses. We refer to these model variants as (p1, p2, shuff, random) respectively. The base model (p1) used first half of observed spiking activity for estimating weights and evaluated model performance on the remaining half, model (p2) used the second half of spiking activity to estimate model weights and evaluate performance on the first half, model (shuff) used a shuffled version of binned activity in which the stimulus labels were independently permuted for each neuron while retaining the within trial bins and model (random) used a shuffled version of binned activity in which both stimulus and within trial bins were independently permuted for each neuron.

### A2.3. Computation of subspace angles

For every pair $(M_1, M_2)$ of interaction matrices between pairs of visual cortical areas, we quantified the degree of alignment between corresponding low-dimensional subspaces using subspace angles (also called principal angles between subspaces). Principal angles $\theta_i(i = 1, 2, \cdots, q)$ between two matrices $U_{n \times p}$ and $V_{n \times q}$ with $p > q$ are defined (Björck and Golub, 1973; Knyazev and Argentati, 2002) by $\cos \theta_k = \max_{u \in U} \max_{v \in V} u_k^T v_k$. To standardize for differences in numbers of recorded neurons between source and target (and thereby the dimensionality of the estimated interaction matrices), we first used singular value decomposition (SVD) to reconstruct 10-dimensional subspaces $(M_1', M_2')$ using the first 10 respective singular vectors. We then calculated the angles between each singular vector of respectively $M_1$ $(M_2)$ and the corresponding 10-dimensional subspace $M_2'$ $(M_1')$ respectively. The average of these two angles was used to quantify the subspace angle between $M_1$ and $M_2$. Directly computing the subspace angles between $(M_1', M_2')$ did not affect our conclusions (Appendix A5.2). Subspace angles have been used in (Bondanelli et al., 2021) to show that OFF responses in auditory cortex to different stimuli lie mostly in orthogonal subspaces.

## Appendix A3. Experimental data

We selected neurons based on three quality metrics which ensured that units were relatively stable across the entirety of each session ($\approx$ 90 minutes) and had low levels of spike train contamination (see Appendix A2). We chose 19 sessions that contained at least 10 units in five out of six visual cortical areas. This gave us a total of 5686 units across the 19 chosen sessions. We provide a breakdown of the number of units per visual area for each of the 19 sessions analyzed in Figure A1.

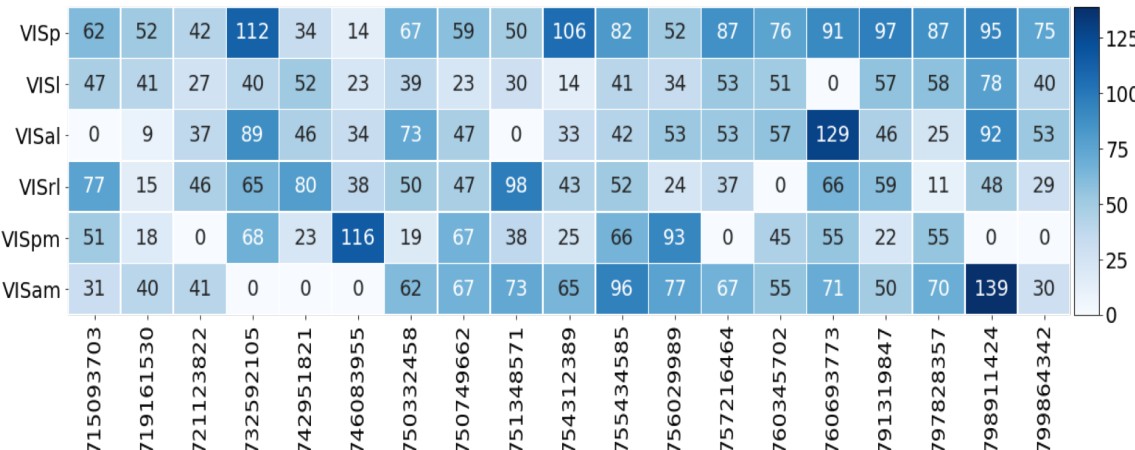

Figure A1: Number of units per visual area for each session analyzed after checking quality metrics (see Appendix A2).

## Appendix A4. Model for interactions

In our model, the instantaneous firing rate $y_i(t)$ of an individual neuron was predicted in time-bins of size 10 ms as the weighted sum of a peer-prediction term and a stimulus-dependent

term as follows,

$$\hat{y}_i(t) = \sum_j w_{ij}y_j(t - D) + \gamma \sum_\beta w_{i\beta} \int dx \int dy\, k_\beta(x, y)S(x, y, t) \tag{A2}$$

where $k_\beta(x, y)$ represents a basis set of 120 gabor filters and $S(x, y, t)$ represents the static grating stimulus. Gabor filters were parameterized using static grating parameters (6 orientations, 5 spatial frequencies, 4 phases) used in the experiments and $D$ represents a delay that allows us to include the spiking history of individual neurons. We chose $D = 10$ ms for our model.

Figure A2 shows a matrix of coefficients $w_{i\beta}$ from the 120 basis gabor filters on to all visual cortical units from an example session.

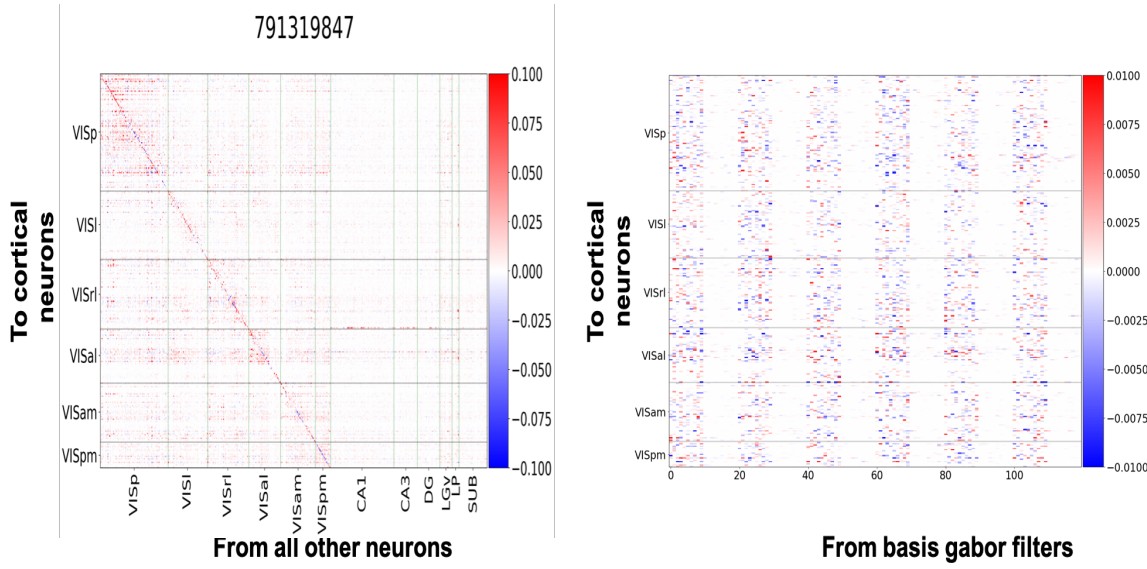

Figure A2: Coefficients for peer coupling and stimulus contribution for one example session. Left: Peer coupling coefficients from units in all regions (columns) on to units in all visual cortical regions (rows). Vertical and horizontal lines demarcate units within a given source and target region. Right: Stimulus coupling coefficients from bank of 120 basis gabor filters (columns) on to units in all visual cortical regions (rows). Horizontal lines demarcate units within a given target region. Colorbar scale has been adjusted for clarity. Each block of 20 columns corresponds to one orientation and each such block is sorted by spatial frequency from low to high. Lower spatial frequencies lead to larger coefficients in the model.

To standardize for differences in numbers of recorded neurons between source and target (and thereby the dimensionality of the estimated interaction matrices) before computing subspace angles, we first used singular value decomposition (SVD) to reconstruct 10-dimensional subspaces $(M_1', M_2')$ using the first 10 respective singular vectors. Figure A3 shows a plot of the cumulative variance explained in an interaction matrix as a function of the number of singular vectors for VISp-VISp, VISp-VISl and VISp-VISam interactions averaged over all 19 sessions. We are able to capture atleast 75% of the variance in these interaction matrices with $N = 10$ singular vectors.

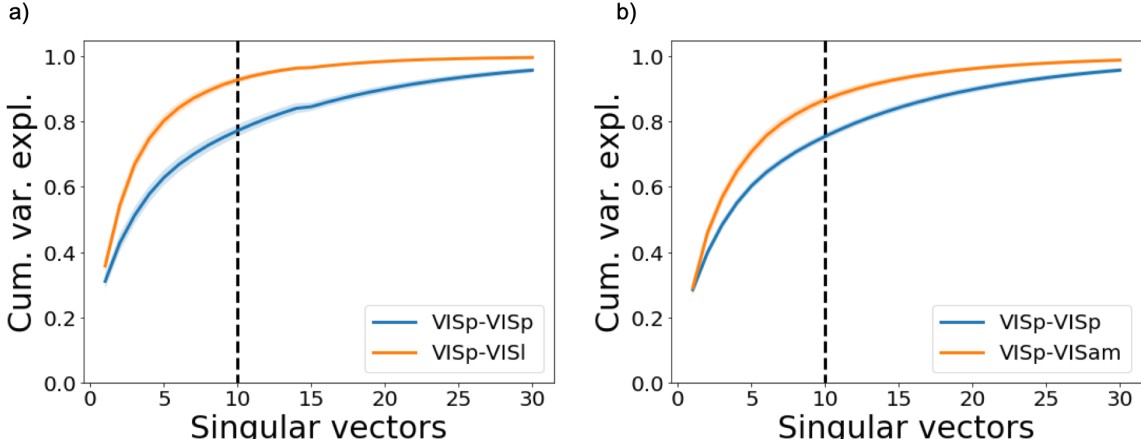

Figure A3: Energy/cumulative variance explained with increasing number of singular vectors of interaction matrices averaged across all sessions. a) Interaction matrices VISp-VISp (blue) and VISp-VISl (orange), b) Interaction matrices VISp-VISp (blue; same curve as in a) and VISp-VISam (orange). Vertical dashed lines correspond to number of singular vectors we used for estimating subspace angles. Our choice of $N = 10$ singular vectors captures at least 75% of variance in the interaction matrices.

### A4.1. Total amount of compute and type of resources

Models parameters for each unit were estimated with parallelization using nodes on our internal cluster. For each unit, model fitting took $\approx 5$ minutes on average and used 10GB of memory. Across all analyses and model variants presented, we had to run approximately 100,000 jobs.

## Appendix A5. Cortical areas interact via distinct subspaces

Figure A4 shows that internal-external first and second angles $\theta(S, S, T)$ are significantly higher than those obtained after shuffling the activities (see Appendix A2, model variant shuff. Wilcoxon signed-rank test pvals ($9.0 \times 10^{-15}, 5.6 \times 10^{-12}, 2.8 \times 10^{-13}, 3.9 \times 10^{-14}, 1.5 \times 10^{-11}, 5.6 \times 10^{-13}$) respectively for first angles in Fig. S4e top row and ($3.9 \times 10^{-11}, 5.2 \times 10^{-8}, 2.6 \times 10^{-7}, 0.0001, 0.002, 5.3 \times 10^{-6}$) for second angles seen in Fig.S4e bottom row for six source visual cortical areas respectively).

Angles $\theta(S, T_1, T_2)$ between external-external interactions are also higher than shuffle controls for the first angle as can be seen in Figure A5 ((Wilcoxon signed-rank test; pvals : ($1.9 \times 10^{-38}, 6.3 \times 10^{-22}, 1.2 \times 10^{-31}, 1.3 \times 10^{-31}, 3.7 \times 10^{-25}, 1.9 \times 10^{-36}$) respectively for first angles (Fig. S5c)).

However, the second angles in this case are not significantly higher than shuffle controls except for source area VISp.((Wilcoxon signed-rank test; pvals : ($0.003, 0.997, 0.878, 0.999, 0.999, 0.999$) for second angles (Fig. S5d) for six source visual cortical areas respectively).

### A5.1. Subspaces exhibit greater alignment during the wave of activity elicited by flashed stimuli

We have shown (Figure 4) that subspaces exhibit greater alignment during the wave of activity elicited by flashed stimuli between 50-150 ms after stimulus onset. An important question is if this observed change is the result of changes in firing rates within these intervals.

To address this, we used computed subspace angles from interaction matrices obtained after shuffling activities (see Appendix A2, model variant shuff). The same shuffle, which preserves the temporal structure of the average firing rate of the neurons was used, but it was applied to neuronal activities separated into five 50ms time intervals. We applied the same method to compute the subspace angles and their relative alignment for the shuffled data, and we subtracted the shuffle control from all the angles computed.

We find that the results are unchanged when the shuffled angles are subtracted, with significant alignment between the inter- and intra-areal subspaces 50-150 ms after the stimulus onset compared to 0-50ms (Figure A6) ((Benjamini-Hochberg FDR-corrected p-values for Wilcoxon signed-rank test comparing average shuffle-subtracted angles for epochs 2-5 with epoch 1; first angle p-values : $(1.35 \times 10^{-13}, 3.97 \times 10^{-12}, 8.60 \times 10^{-8}, 7.50 \times 10^{-3})$, second angle p-values : $(2.18 \times 10^{-14}, 1.81 \times 10^{-12}, 1.54 \times 10^{-2}, 9.86 \times 10^{-2})$))

### A5.2. Direct computation of subspace angles

Directly computing the subspace angles between the matrices $(M_1', M_2')$ reconstructed using the first 10 singular vectors of $(M_1, M_2)$ did not affect our conclusions. Figure A7 (panels c) and d)) show results for both internal-external and external-external angles combining the results seen in Figures A4 and A5. Panel b) shows that angles obtained are higher than shuffle controls.

### A5.3. Subspace angles from interaction matrices obtained using approach of Semedo et al. (2019)

We also computed subspace angles after estimating interaction matrices using the approach of Semedo et al. (2019). We first computed the trial-to-trial fluctuations in response to static gratings for all sessions. We used these to relate activity fluctuations in a target area to those in a source area using reduced rank regression (RRR). To do so, we used the MATLAB code made available by the authors at (https://github.com/joao-semedo/communication-subspace). We used the mapping matrix returned by their code for given source and target variables as the estimated interaction matrices from which we computed the subspace angles.

Figure A8 shows subspace angles (averaged across all sessions) from source area S=VISp to pairs of target areas (VISp, VISl), (VISp, VISam) and (VISl, VISam). It can be seen that the internal-external angles are larger than the external-external angles, substantiating our result in Figure 3 that internal-external interactions are less closely aligned than external-external interactions.

## Appendix A6. PredNet analysis

We have shown that intra-areal interaction subspaces in mouse visual cortex are not well aligned with the inter-areal interaction subspaces (Figure 3). Do similar patterns of interactions exist within task-trained artificial neural networks? To allow for such a possibility, the network is likely to require interactions between neurons within an area. Therefore we analyzed interactions between the representation modules in PredNet( (Lotter et al., 2016, 2020)) - a deep convolutional recurrent neural network that was trained for next-frame video prediction. PredNet's architecture was inspired by principles of predictive coding from neuroscience (Rao and Ballard, 1999), but it includes both feedback from higher areas as well as lateral connection on top of the feedforward connections.

We used the publicly available code-base provided by the authors of PredNet ([https://github.com/coxlab/prednet](https://github.com/coxlab/prednet)). We 'probed' the activations of a subset of neurons within representation layers/modules $R_1$, $R_3$ and $R_3$ in PredNet in response to video sequences provided as test set by the authors of PredNet. The test sets consists of 83 video sequences of 10 frames each. The representation are the layers that have explicit recurrent connections. The depths of these layers are $R_1 = 48$, $R_2 = 96$ and $R_3 = 192$ respectively. We chose a subset of 72 neurons within each of these 3 layers by considering a 3 x 3 spatial region in the center and every 6th, 12th or 24th slices along the channel axes in these layers. We evaluated models both without and with self-interactions between units with a delay and obtained qualitatively similar results with both. For brevity, we report results here only for the model without delayed self-interactions between units. To mimic different sessions/mice as in the biological data, we split the activity matrix into 10 chunks of equal sizes and created a training set by randomly choosing 5 chunks and using the remaining chunks for testing. We repeated this process for $n = 5$ random seeds from 0 to 4 inclusive and treated each such instance as a model arising from a different session.

We used the peer coupling model to estimate functional interactions among the chosen neurons within and between these layers for the different model variants. Figure A9 shows estimated interaction matrices from source module $R_2$ to target modules $R_1$ and $R_2$.

In contrast to the biological network, our analysis did not result in significant differences between internal-external and external-external interactions within PredNet (Figure A10 ((Benjamini-Hochberg FDR-corrected p-values for the Mann-Whitney U test for the three representation modules respectively: $(0.613, 0.860, 0.275)$ for the first angles and $(0.987, 0.306, 0.404)$ for the second angles))

To investigate if noise influences the geometry of interactions, we injected a small amount of noise in to the activities of chosen units in each module and then re-computed subspace angles after estimating the respective interaction matrices. For each module, we added Gaussian white noise to all activities within that module with zero mean and standard deviation $\sigma_s = 0.01 \times \sigma$, where $\sigma$ is the standard deviation of activities of the chosen 72 units within that module. Again, we did not find significant differences in internal-external and external-external interactions between PredNet modules (Figure A11) (Benjamini-Hochberg FDR-corrected p-values for the Mann-Whitney U test for the three representation modules respectively: $(0.403, 0.370, 0.879)$ for the first angles and $(0.403, 0.306, 0.840)$ for the second angles)).

In PredNet, the absence of significant differences in internal-external and external-external interactions is similar to what is observed in mouse visual cortical areas 50-100 ms after stimulus onset when the misalignment between subspaces disappears (Figure 4 and Figure A6). There are many possible reasons for the observed differences in interaction patterns in PredNet and visual cortical areas and we can only speculate. One possibility is that visual cortical areas can dynamically rotate the interaction subspaces and reduce the misalignment only when task relevant, while in the task-driven PredNet network the representation modules with recurrent interactions do not become misaligned.

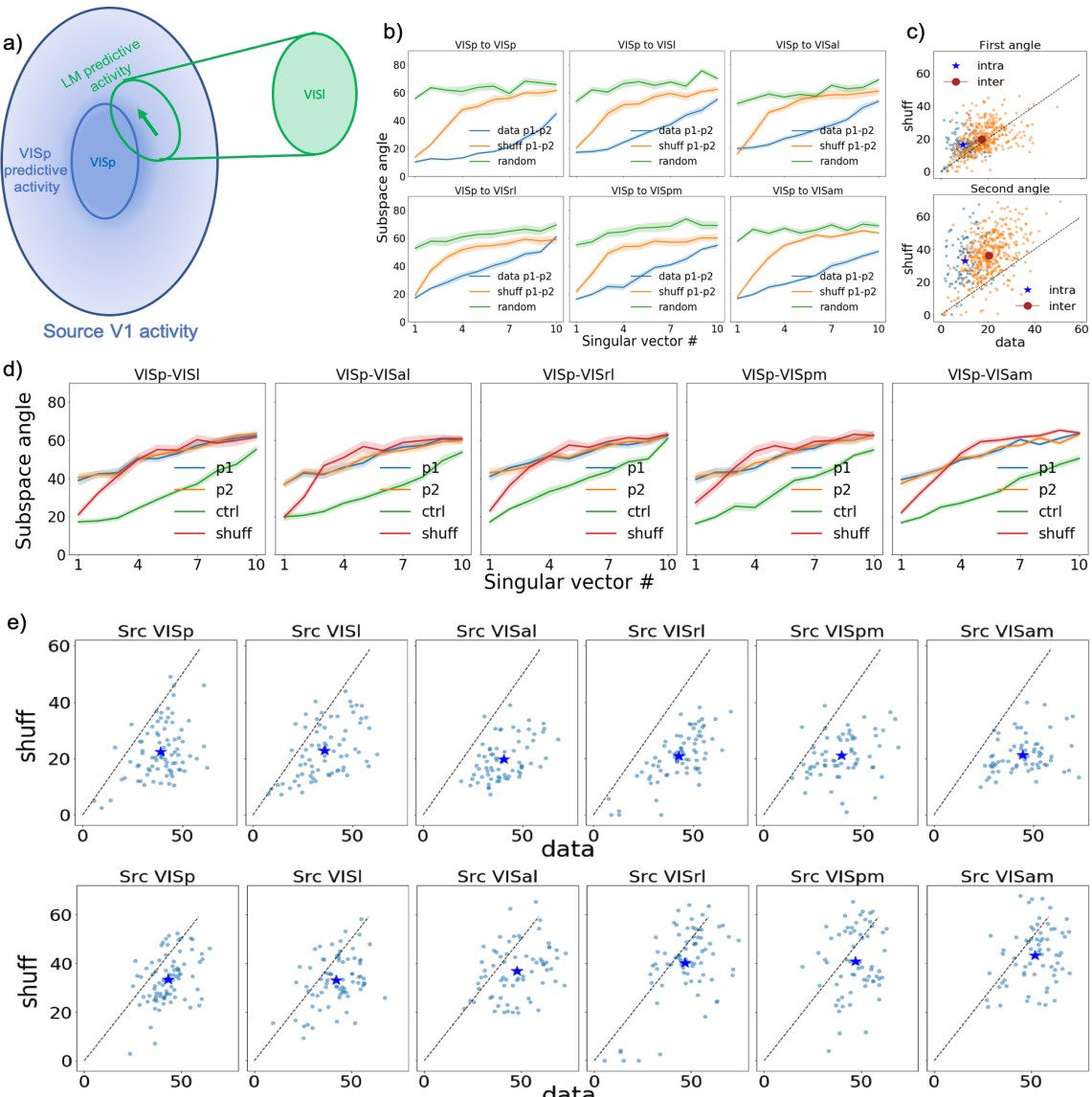

Figure A4: Distinct subspaces of source areas mediate intra- and inter-areal interactions. a) Schematic of the idea showing that distinct subspaces in VISp mediate interactions within VISp and outside with VISl. b) Subspace angles (see Appendix A2) provide a measure of how distinct interaction subspaces between pairs of areas are aligned. Subspace angles for interaction matrices from VISp to six target regions estimated using first and second halves of activity for original data (blue), shuffled data (orange) and a fully randomized condition (see Appendix A2). Solid lines represent session averages; shaded regions show std error of mean (sem). c) Scatter plots of first (top) and second (bottom) subspace angles from original data (horizontal axis) vs shuffled data (vertical axis) for intra-areal (blue) and inter-areal (orange) interactions. Angles from original data are significantly lower than those obtained using shuffled data (Wilcoxon signed-rank test; pvals : $(1.8 \times 10^{-13}, 3.7 \times 10^{-9}, 4.4 \times 10^{-18}, 7.3 \times 10^{-67})$ respectively for first intra-, first inter-, second intra- and second inter-areal interaction angles.) d) Subspace angles from a fixed source region (VISp shown here) to (VISp, T) where T is one of five higher visual areas [T : VISl, VISal, VISrl, VISpm, VISam] for model variants p1 (blue) and p2 (orange). First few angles are significantly higher than inter-areal interaction angles (green) estimated from activities in different stimulus blocks (blue lines in Figure 2(b)) as well the shuffle control (red). e,f) Scatter plots of first and second subspace angles separately for each session and source-target combination from original data (horizontal axis) vs shuffled data (vertical axis). The markers show the mean subspace angles.

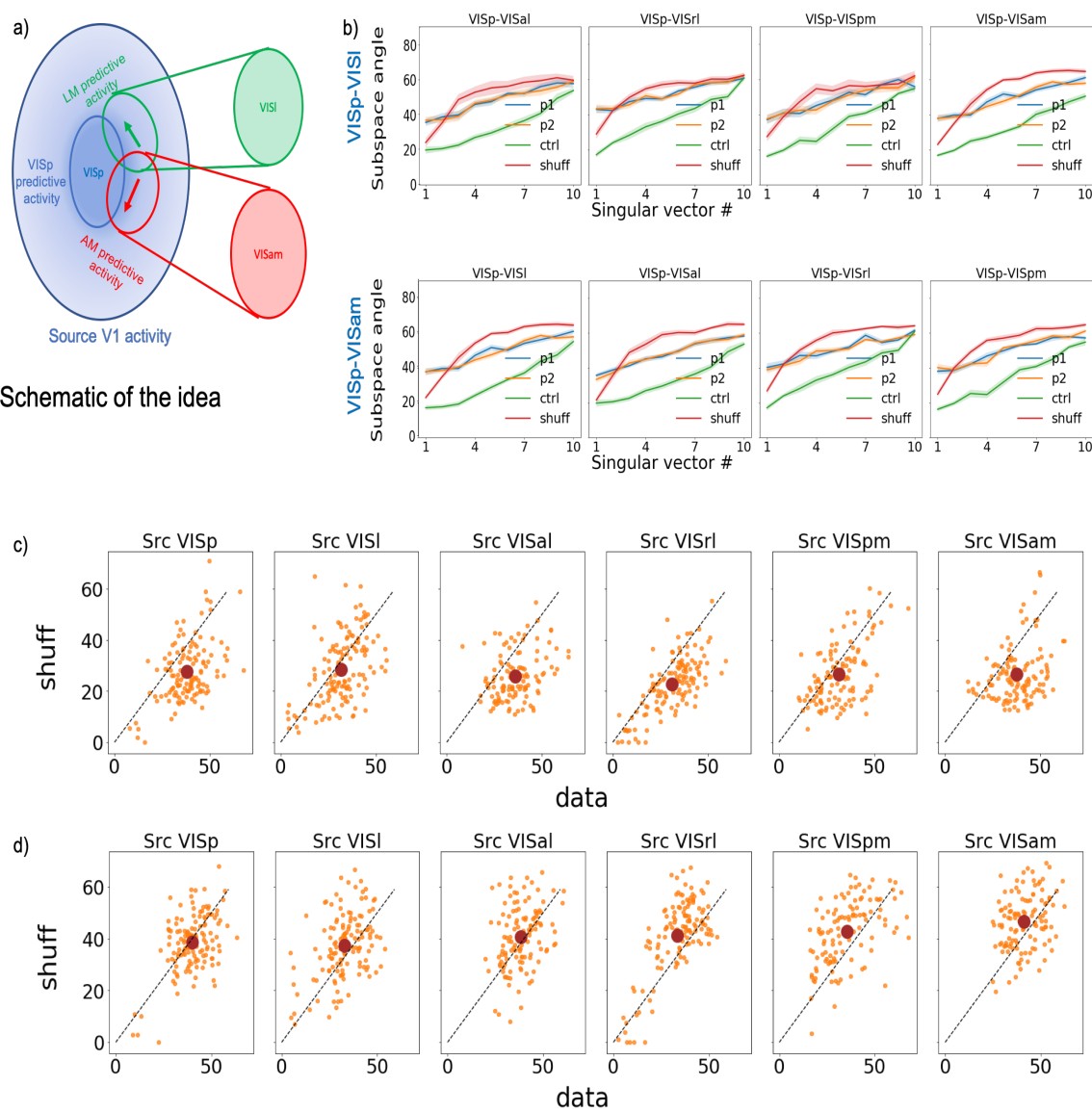

Figure A5: Distinct subspaces of source areas mediate interactions with external target areas. a) Schematic of the idea showing that distinct subspaces in VISp mediate interactions outside with VISl and VISam. b) Subspace angles between VISp-VISl (top) and VISp-VISam (bottom) subspaces and remaining four external target subspaces estimated using first (blue) and second (orange) halves of activity for original data, shuffled data (red) and angles (green) estimated from activities in different stimulus blocks (blue lines in Figure 2(b) and Figure A4. Solid lines represent session averages; shaded regions show std error of mean (sem). c, d) Scatter plots of first and second subspace angles separately for each session and source-target combination from original data (horizontal axis) vs shuffled data (vertical axis). The markers show the mean subspace angles.

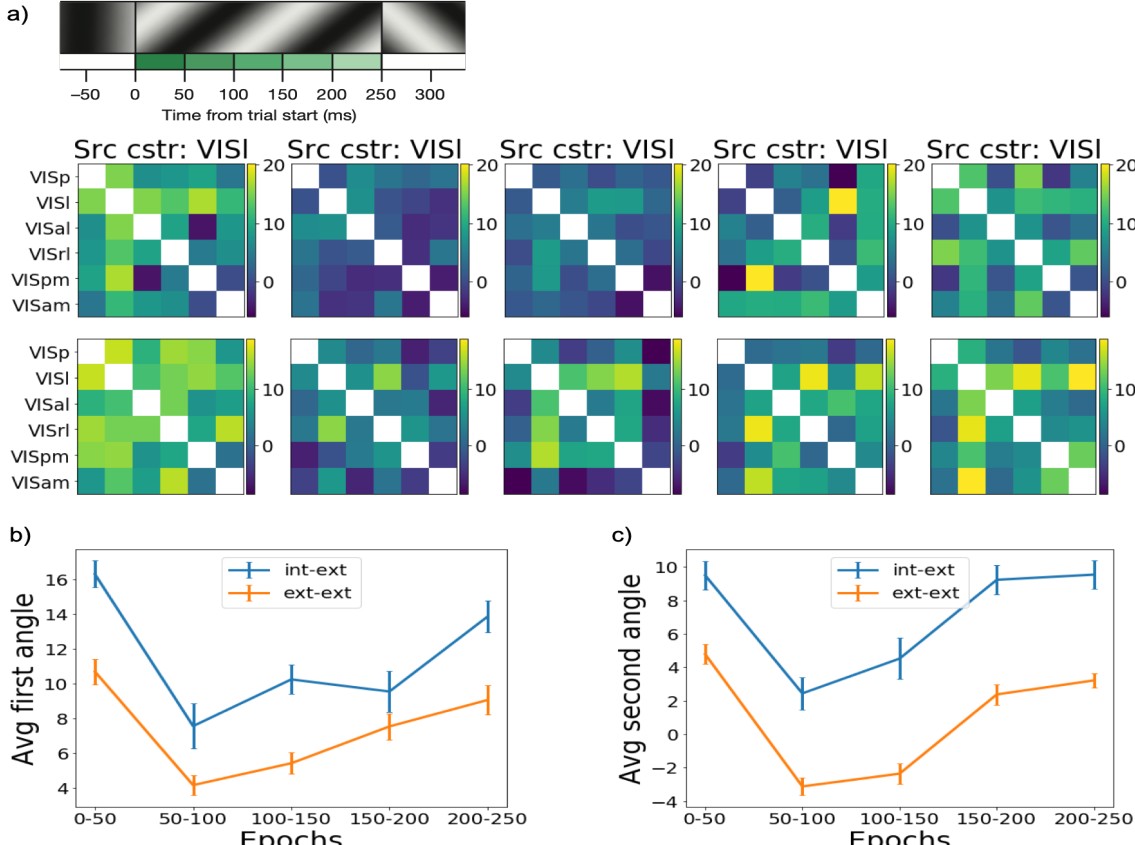

Figure A6: Subspaces rotate within single trials and become more aligned after stimulus onset and results remain unchanged even after shuffle-subtraction (see text for details). a) Top: Schematic showing division of 50ms epochs within a single trial from stimulus onset. Bottom first row: Heatmap of shuffle-subtracted first angle from a source cortical area (VISl shown here) to all possible target cortical area pairs for five successive 50 ms epochs (columns) within 250 ms trials. Bottom second row: Shuffle-subtracted second angles. Angles were computed by estimating interaction matrices using activity within each 50 ms epoch. Note that although subspaces become more aligned, internal-external subspace pairs are still more orthogonal than external-external subspace pairs within each epoch. b) Average of all combinations of internal-external (blue) and external-external (orange) shuffle-subtracted first angles for the five 50 ms epochs showing subspace alignment between 50-150 ms after stimulus onset. c) Same as in b) but for second angles.

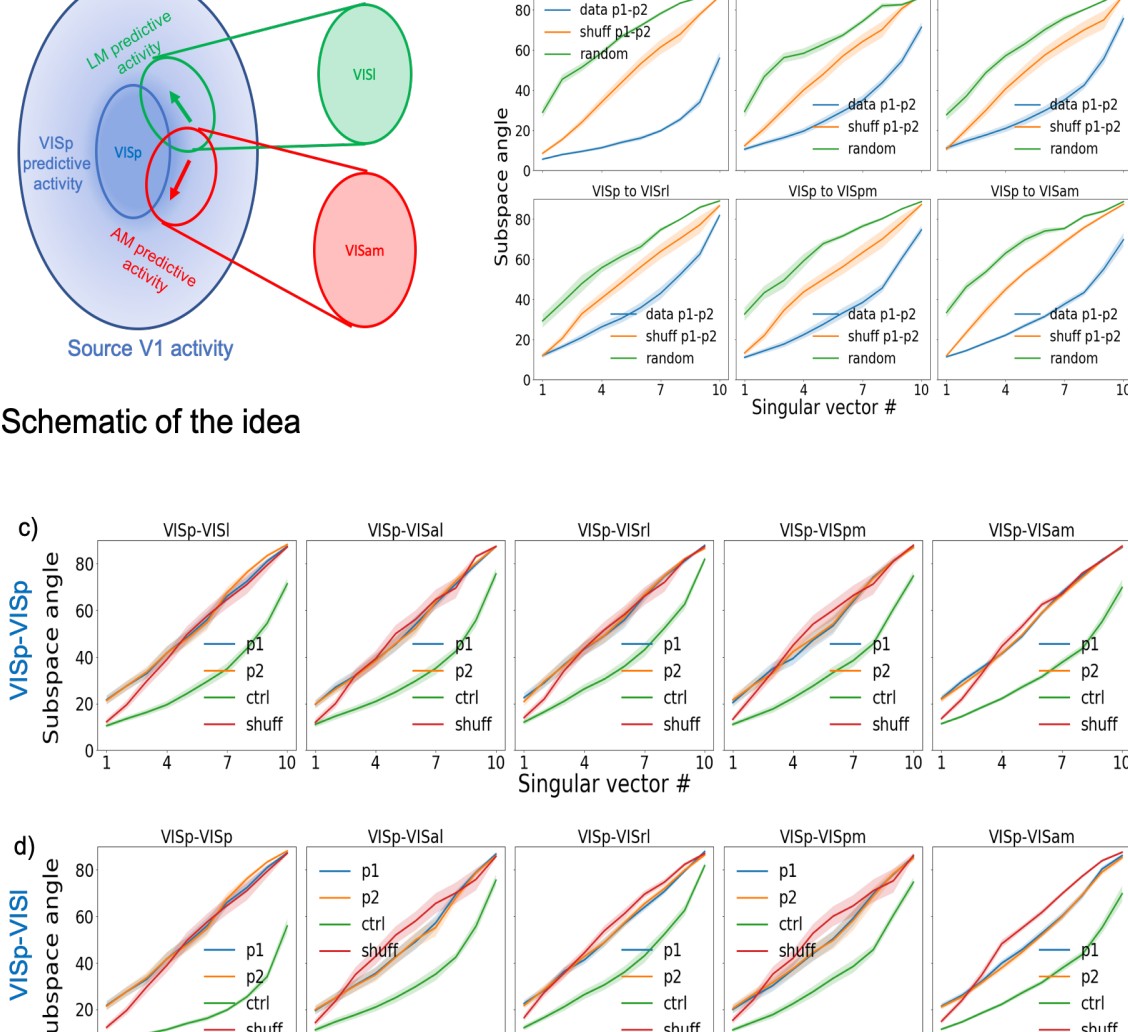

Schematic of the idea

Figure A7: Subspace angles estimated directly from reconstructed interaction matrices. a) Schematic of the idea showing that distinct subspaces in VISp mediate interactions within VISp and outside with VISl. b) Subspace angles for interaction matrices from VISp to six target regions estimated using first and second halves of activity for original data (blue), shuffled data (orange) and a fully randomized condition (see Appendix A2). Solid lines represent session averages; shaded regions show std error of mean (sem). c) Subspace angles from a fixed source region (VISp shown here) to (VISp, T) where T is one of five higher visual areas [T : VISl, VISal, VISrl, VISpm, VISam] for model variants p1 (blue) and p2 (orange). First few angles are significantly higher than inter-areal interaction angles (green) estimated from activities in different stimulus blocks (blue lines in panel b) as well the shuffle control (red). d) Subspace angles between VISp-VISl and (VISp, VISp) and remaining four external target subspaces estimated using first (blue) and second (orange) halves of activity for original data, shuffled data (red) and angles (green) estimated from activities in different stimulus blocks (blue lines in Figure 2(b) and Figure A4. Solid lines represent session averages; shaded regions show std error of mean (sem).

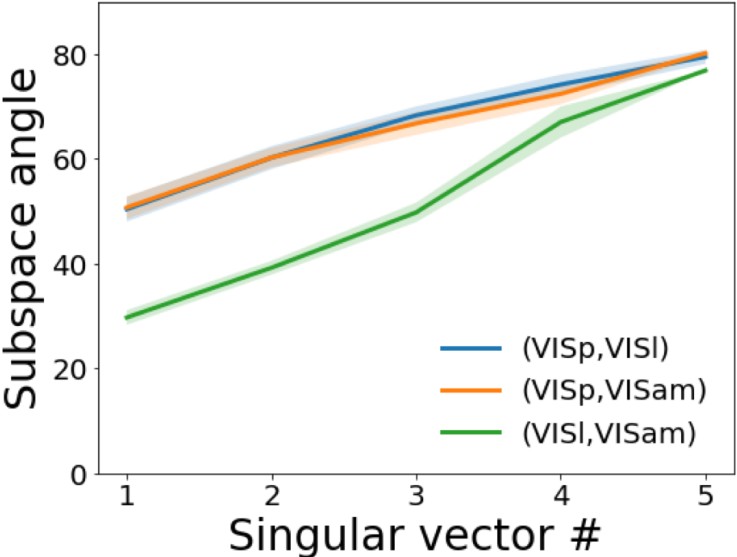

Figure A8: Subspace angles between interaction matrices estimated using reduced rank regression as outlined in Semedo et al. (2019) (see text for details) from source area S=VISp to three pairs of target areas (VISp, VISl), (VISp, VISam) and (VISl, VISam). Solid lines represent average across all sessions and shaded lines represent standard error of mean (sem). Internal-external interactions are less closely aligned than external-external interactions.

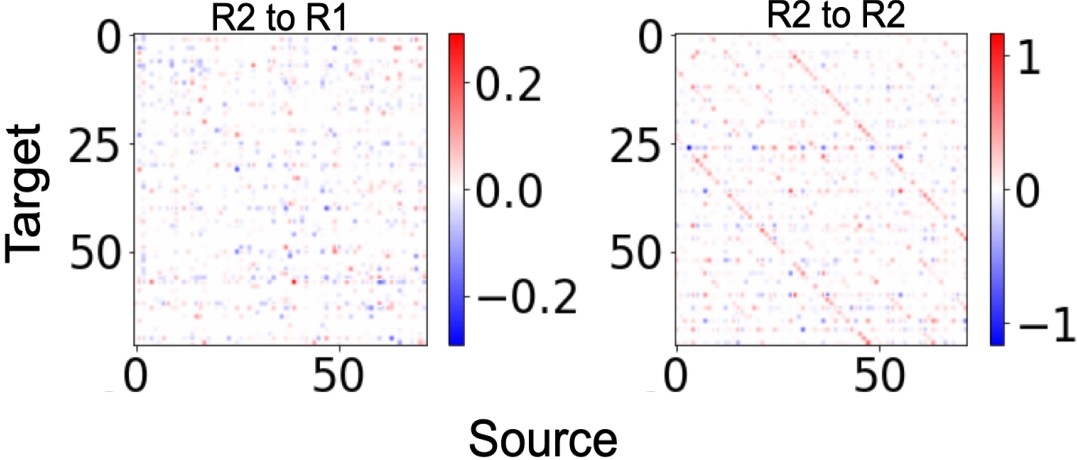

Figure A9: Functional interactions in PredNet obtained with our model for source module $R_2$ to target modules $R_1$ (left) and $R_2$ (right) respectively. Model does not include delayed self-interactions for units (see text for details).

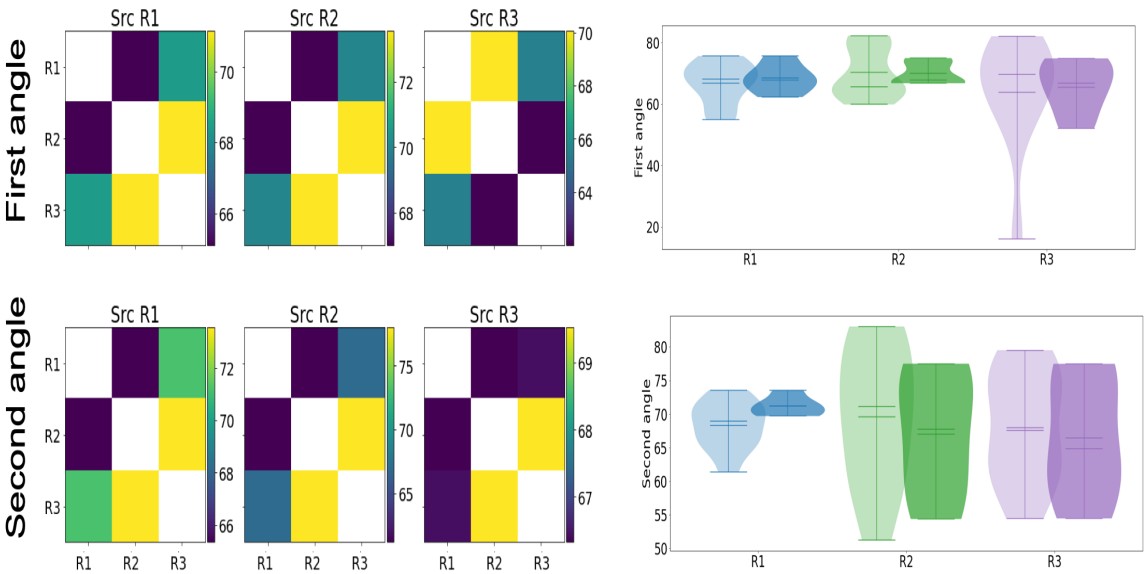

Figure A10: Internal-external and external-external interactions within PredNet are not significantly different from each other. Left: Heatmaps show average first angle (top row) and second angle (bottom row) across different random seeds that act as a proxy for different sessions/mice (see text for details). Right: Distributions of internal-external and external-external angles (top; first angle, bottom; second angle). Each pair of violinplots with different shades of the same color represents internal-external (light shade) and external-external (dark shade) angles respectively.

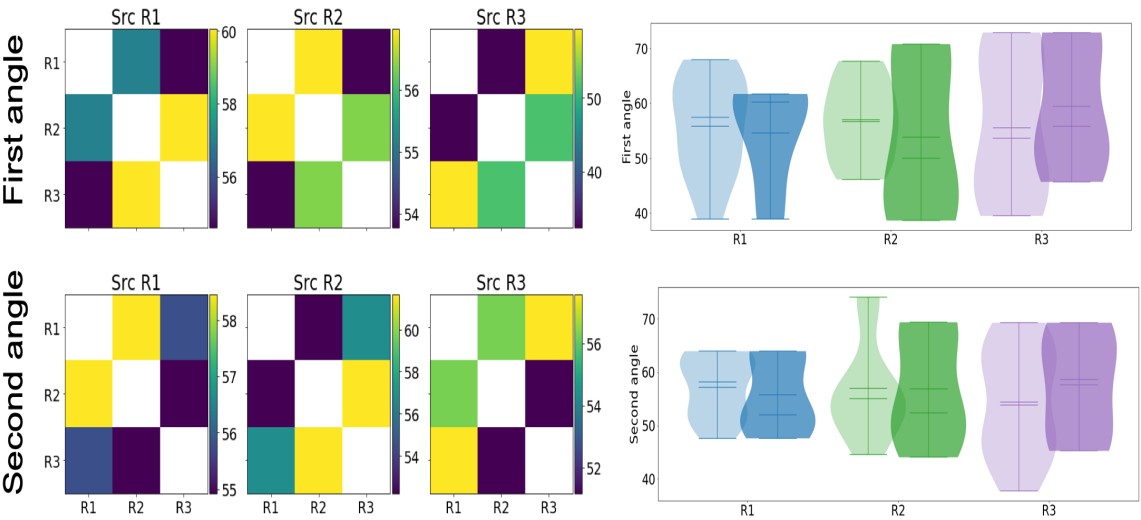

Figure A11: Internal-external and external-external interactions within PredNet are not significantly different from each other even after adding Gaussian white noise to chosen unit activities within respective modules (see text for details). Left: Heatmaps show average first angle (top row) and second angle (bottom row) across different random seeds that act as a proxy for different sessions/mice (see text for details). Right: Distributions of internal-external and external-external angles (top; first angle, bottom; second angle). Each pair of violinplots with different shades of the same color represents internal-external (light shade) and external-external (dark shade) angles respectively.

