# OpenReview forum: "Geometry of inter-areal interactions in mouse visual cortex"
_NeurIPS.cc/2022/Workshop/NeurReps — NeurReps 2022 Poster_

### Official Review · Reviewer_i1wd · 2022-10-12
**Generalization of previous findings of communication subspaces to Neuropixel dataset**

**Confidence:** 4
**Soundness:** 3
**Presentation:** 2
**Contribution:** 2
**Overall Rating:** 6

**Summary:**

Authors analyzed the Allen Brain Observatory Neuropixel dataset to investigate the functional interactions of multiple areas of the visual cortex. Using a linear model (variant of Semedo et al 2019), they quantified the angles between communications subspaces, i.e. angle(S,T1,T2) where S is source area and T1 and T2 are target areas. They found that intra-areal interaction subspaces are less aligned than inter-areal interaction subspaces, i.e., the intra-areal angles are larger than the inter-areal angles. In addition, they demosntrated that the alignment of subspaces can dynamically change in time.


**Questions:**

* Here I use the notation (S,T1,T2). In Fig2(b), S=VISp but what are T1 and T2? From the titles of individual panels in Fig2(b), can we read off what T1 and T2 are? In the legend of Fig2(d), it stated that the blues lines in Fig2(b) are inter-areal interaction angles. So, I'd assume T1 and T2 are different from S=VISp. Please clarify what are T1,T2 in Fig2(b) and why it depicts inter-areal interaction.

* Fig 3(e) contains the main result of the paper. It's not shown.


**Limitations:**

* What are the implications of intra-areal communication subspaces being less aligned than inter-areal communication subspaces? It would be good to discuss the significance of the findings.


**Recommended Decision:**

2: Borderline

**Relevance:**

3: Solid fit

**Strengths And Weaknesses:**

# Originality
First, the use of Allen Brain Observatory Neuropixel data to analyze the interactions of multiple visual areas is a novel aspect of this study. The analysis method is standard. Second, the finding that intra-area communication subspace is less aligned than inter-area communication subspace is a generalization of previous findings to multiple visual areas, so I think this would be an observation useful to the neuroscience community.
# Quality
This study employed techcnically sound analysis method based on Semedo et al 2019.
# Clarity
I had major issues with understanding the results that support the main claims. This is the part that needs major improvement.
* Authors use two different terminologies to characterize communication subspaces: (1) intra-areal vs. inter-areal and (2) internal-external vs. external-external. It is unclear if they refer to the same concepts or not.
* Some figures referred to in the main text were missing. For example, Fig3(e) discussed in the first paragraph of page7 is NOT included in the manuscript. I think this figure was supposed to show their main finding.
# Significance
The results are useful to the neuroscience community as they extend previous findings to ABO dataset, but it is not clear to me if the results have significant impact.


**Submission Track:**

Proceedings Paper (9 Page)

---

### Official Review · Reviewer_zHdR · 2022-10-14
**Thorough analysis, but challenging to interpret**

**Confidence:** 3
**Soundness:** 3
**Presentation:** 2
**Contribution:** 2
**Overall Rating:** 5

**Summary:**

The authors analyze a well-known public dataset from the Allen Brain Observatory and analyze inter-area functional interactions in terms of the "communication subspace" ideas developed by Semedo et al. (2019). Their method is simple, but interesting. The results appear largely confirmatory of Semedo et al.'s observations but in a different model system (mice instead of nonhuman primates) and across multiple brain areas. A potentially novel result is that the geometry of inter-area communication subspaces dynamically changes over the course of each trial. The paper is very long and developed. I was not able to identify any serious technical errors.

**Questions:**

I am combining questions and some comments here:

- On pg 3, the authors say they used a generalized linear model, but from looking from the appendix it isn't exactly clear whether this was a GLM with a Poisson noise model or just ordinary least squares regression. This should be clarified on a technical level.

- Figure 2 should explain clearly what is the function $\theta(a, b, c)$ - perhaps similar to the diagrams in Semedo et al.

- Related to above, the subspace angles are typically defined between a pair of matrices so, I don't understand why $\theta(\cdot)$ takes three inputs instead of two.

- In Figure 2b the first principle angle is the same for shuffle and data. The authors briefly mention it but I think it deserves more unpacking; at least I don’t understand how to interpret this.

- Related to above, why do you only study only the top two subspace angles? It feels necessary to compare all ten subspace angles to get the full picture.

- In Figure 3, colormap scales are different across heatmaps, making comparisons very challenging.

- Should the heatmaps in Figure 3 and 4 be normalized to a shuffle control somehow?

**Limitations:**

The authors have done a reasonable job of identifying limitations. My main concern is that I'm not sure "what it all means", especially in light of the challenges in interpreting regression coefficients that I highlighted in my comments above and are discussed in detail by Kriegeskorte & Douglas (2019).

**Recommended Decision:**

2: Borderline

**Relevance:**

3: Solid fit

**Strengths And Weaknesses:**

The biggest strength of this paper is its thoroughness and lengthy appendix which are quite long for a workshop submission. It also uses public data so the results should be easy for others to replicate.

The writing and figures are difficult to parse in places, and the take-home messages of their analysis are sometimes challenging to identify. I have listed several questions in the next section below to highlight some areas where I got confused. In several places in the main text there are details that felt as if they could be relegated to the appendix, while at the same time important details were put in the appendix that made it difficult to read through the main text in a straightforward manner.

Interpreting regression coefficients in complex, multivariate datasets is a very challenging and even counter-intuitive endeavor. See Kriegeskorte & Douglas (2019) "Interpreting encoding and decoding models" for detailed explanation of what I mean. The authors should be commended for setting out to a challenging, but important and meaningful study. I do worry that some of the subtle details outlined in the review above are not accounted for and that greater care needs to be taken in interpreting these results for readers.

For example, on pg 6, it is claimed that subspaces "M1 and M2 correspond to interactions between source area S and external target areas T1 and T2". But it is not clear to me what "interactions" means in this context and Fig 1 in Kriegeskorte & Douglas cited above shows that such "interactions" are a complex mixture of signal and noise correlations. Further, it would be useful to explain how the subspace identified by regression is different than the subspace defined by different methods like, e.g., principal components analysis.

**Submission Track:**

Proceedings Paper (9 Page)

---

### Official Review · Reviewer_gMrs · 2022-10-14
**Review of geometry of inter-areal interactions in mouse visual cortex**

**Confidence:** 4
**Soundness:** 3
**Presentation:** 3
**Contribution:** 3
**Overall Rating:** 7

**Summary:**

The authors examine interactions between and within visual areas. They used open source Neuropixel spiking data obtained by Allen Brain Institute from mice who were headfixed and observing a series of static grated stimuli (Figure 1). Next, they refined most of their analyzes to interactions within areas of visual cortex. They choose 6 areas and recording sessions that contained a minimum of 10 units (with a max of 129). Next, they fit a peer-prediction generalized linear model (L1 regularization) to predict the neural activity either within a region or to another region: source  target or source  source (self). These fits resulted in an interaction matrix (e.g., M1 or M2). Throughout the paper, they look at how one source predicts activity in two areas (e.g., I will write S | S, T to mean source is predicting activity in itself (S) and another target brain area (T). First, they look at how sources influence themselves vs. a target region (Figure 2). Given the interaction matrices M1 and M2, they calculate the Principal Angles between the subspaces. Most of the paper shows the resulting first and second angles. In Figure 2, they show that intra-regional angles (S | S,S) are smaller than the inter-regional angles (S | S,T). Next, in Figure 3, they compare these results to how a source activity (S) is able to predict activity in other target regions (S | T1, T2). They show that the angles between a source’s ability to predict itself and it’s ability to predict another target (S | S,T) is larger that the angles between a sources ability to predict one target or another target. Essentially, this means that the way a source projects into two regions is more similar than how a source projects to one region, compared to how it projects to itself. Finally, in Figure 4, they examine how the alignment of these subspace angle between regions evolves over time. Despite changing within trial, the angles themselves are consistent across trials.

Overall, they conclude that subspaces are different between and within areas and that within area interactions are less aligned compared to between area interactions. While they find that subspace interactions evolve over time, the mean similarity between subspaces is stable across stimulus presentation.


**Questions:**

It would help if the authors clarify how the L1 regularization is set. How is significance determined to create the M1/2 interaction subspaces?

It would help if the authors either clarify / justify why they use only 1-2 Principal Angles rather than taking the sum or average of Principal Angles, which would give a more complete picture?

It would help if the authors use different colors for different analyses. All the colors are similar in Figure 2 (and others), yet they refer to different interactions, models etc and this makes for a confusing read.

It would help if the authors label the figures with the full set of comparisons being done. For a figure like F2b showing angles between two subspaces M1 and M2, can the authors please list which S-T or S-S interactions were fit? I assume, but I am not sure, the first panel of Figure 2b is VISp- VISp (M1) vs. VISp - VISp (M2) or could be written as (VISp | VISp, VISp). Then the second panel of Figure 2b would be (VISp | VISI, VISp) and so on?

It would help if the authors show / plot visually the mean activity in the subspaces they find. Maybe I missed this…It looks like maybe Figure 1e is this, but averaged across all areas? I would like to see the mean subspace activity per area (S-S) and across areas (S-T). I would like to see this on average compared to the average (S-T) interaction. Furthermore, I would like to see if there is a difference in sparsity between (S-S) interaction matrices and (S-T) interaction matrices.

I understand the need to include as much data as possible in this project, so the authors set the minimum number of neurons needed in a session to 10 units. However, it appears that some sessions had over a hundred neurons. Given that some of those neurons may or may not have been selective to the stimuli, I wonder to what extent a low number of neurons compromised the GLM’s ability to properly fit peer interactions for the <10 unit case? Perhaps I missed this, but should the authors include an analysis showing that the number of units recorded (given that they vary so much here) did not greatly affect their results?


**Limitations:**

The authors discuss several limitations of their work. They suggest that it would be interesting to extend this work to more brain areas and more tasks. They also suggest some interesting further ideas, including combining the analysis of communication subspace (spiking information) with the communication through coherence hypothesis (local field potential/oscillation information) to understand the interaction between these two forms of communication. Finally, the authors state that while they were able to define and characterize these subspaces, they were not able to describe or suggest a mechanism by which they arise, change or are learned in a task dependent manner.

I agree with the authors that those are interesting future directions. I hope they pursue those, but I also wonder if it might be more satisfying and practical for them to finish the work they started here? It seems they have the tools ready to explore what information is actually being communicated through visual cortex and how the representation of the gabor changes along the hierarchy and how that object representation comes together through the action of these subspaces. I also encourage them to address the minor questions/confusions I have previously mentioned.


**Recommended Decision:**

3: Accept

**Relevance:**

3: Solid fit

**Strengths And Weaknesses:**


It is interesting and currently popular approach to examine how information is communicated between brain areas via population subspaces. Indeed, the authors discuss previous related work. It is unclear what open questions they are answering by characterizing the angles of subspace interactions, but I believe the logical flow of their work is potentially sound enough to publish (although I suspect many readers will struggle to understand it).

It might have overall been more interesting if they explored the qualitative nature of information moving through these subspaces. There are countless theories and models of how visual information changes through the visual system’s hierarchy. Given that they are examining how information moves through visual areas, I am surprised they did not test any of these hypotheses. For example, does the visual information become more or less complex through subspace interactions? Which features of the visual object are passed between regions, why and why not? Does this fit with how convolutional neural networks process information. Or do these communication subspaces predict the formation of object representations from the collection of individual features at lower levels of computation?

As it stands now, the paper seems to be a demonstration of how to calculate subspaces and angles between those subspaces. Given that there many interesting follow-up questions to thinking about communication between areas as population subspaces, I do think this paper is an important and relevant contribution to the field. If clarified, I believe others may draw inspiration about how to analyze and think about their own data from this paper. Therefore, I think with some fixes and clarifications, this work should be included in this workshop.


Currently, authors mainly draw “characterizing” conclusions about what they find (i.e., that intra-regional interactions are less aligned than inter-regional interactions). However, I have some concerns about both their methods and interpretation.

Unless I missed it, the authors never explain how the result from fitting their GLM gives them the M1 subspace interaction. I am assuming that the subspace is derived from the weight matrix in the GLM, which is made sparse by the L1 regularization? However, the authors do not say what level of L1 regularization they used or how this may impact their results. I fear that a high level of regularization will emphasize strong interactions within or between regions and ignore weaker (yet still significant) interactions within or between regions. Thus, the authors may wish to consider a metric to set their threshold that maintains interactions that are related to or relevant to coding. In other words, if the subspaces only capture the highest (but not necessarily all the influential connections), they are leaving out a key form of communication.

Second, unless I missed it, the authors do no explain or justify only analyzing the first 2 Principal Angles. Why not take the sum or average of Principal Angles? This might control for an interaction where one area has many diffuse, but overall high, influence on another area compared to an interaction where one area has a few strong, but overall low, influence on another area. Again, it these sums and averaged would be influenced by the level of L1 regularization or forced sparsity in the M1 subspaces.

Finally, I took me forever to wrap my head around this statement: “intra-regional interactions are less aligned than inter-regional interactions.” I’m still not sure I agree with it. At first, I thought the authors meant that the ability of a given region to predict its own activity is less than its ability to predict target regions. But then I realized that Figure 2 and 3 show that (S | S, S) < (S | T,T) < (S |T,S). Overall, I believe this means they found that 1) one area can predict its own activity very well (S-S, intra-area subspace or recurrence) 2) one area sends out information to other areas via subspaces (S-T, inter-area subspaces or projections). Finally, because (S | T, T) < (S | T,S), this means that when comparing across areas (*), you’ll find that there is more similarity in how a bunch of areas receive information from one source (S-T*) compared to the similarity between a region’s recurrence and its projections (S-S vs. S-T*). This essentially implies that areas have subspace for recurrent and another to send information outward and generally they are separate. However, it should be noted the authors did not compare intra-regional interactions or recurrence between areas; in other words, they did not test similarity across recurrences or (S*-S*). This would require comparing the following subspaces M1 (Sa -Sa) and M2 (Sb-Sb). Then the authors could compare the sum of these principal angles with the sum of the principal angles describing inter-regional interactions (S | T*). This would allow them to test whether intra-regional interactions are more or less similar to one another compared to the similarity of inter-regional interactions.


**Submission Track:**

Proceedings Paper (9 Page)

---

### Decision · Program_Chairs · 2022-10-21

Accept (Poster)